**Data Availability Statement:** All relevant data are within the paper and its Supporting Information files.

# Infant and young child feeding practices and its associated factors among mothers of under two years children in a western hilly region of Nepal

Nabin Adhikari[1]◉, Kiran Acharya[2], Dipak Prasad Upadhya[1], Sumita Pathak[3], Sachin Pokharel[4], Pranil Man Singh Pradhan◉[5,6]◉*

1 Central Department of Public Health, Institute of Medicine, Tribhuvan University, Maharajgunj, Kathmandu, Nepal, 2 New ERA, Rudramati Marga, Kalopul, Kathmandu, Nepal, 3 Nursing Department, Kathmandu Model Hospital, Kathmandu, Nepal, 4 La-Grandee International College, Pokhara University, Kaski, Nepal, 5 Department of Community Medicine, Maharajgunj Medical Campus, Institute of Medicine, Tribhuvan University, Maharajgunj, Kathmandu, Nepal, 6 Nepalese Society of Community Medicine, Lalitpur, Nepal

◉ These authors contributed equally to this work.
* pranil.pradhan@gmail.com

## Abstract

Infant and young child feeding is a key area to improve child survival and promote healthy growth and development. Nepal government has developed and implemented different programs to improve infant and young child feeding practice. However, the practice remains poor and is a major cause of malnutrition in Nepal. This study aims to identify infant and young child feeding practices and its associated factors among mothers of children aged less than two years in western hilly region of Nepal. A descriptive cross-sectional study was carried out among 360 mothers of under two years' children in Syangja district. A semi structural questionnaire was used. Data was entered in EpiData and analyzed using IBM SPSS version 21. Descriptive statistics were used to report the feeding practices and other independent variables. Bivariate and multivariate logistic regression model was used to establish the factors associated with infant and young child feeding practices. The prevalence of breastfeeding, timely initiation of breastfeeding, exclusive breastfeeding, timely initiation of complementary feeding, minimum dietary diversity, minimum meal frequency and minimum acceptable diet (MAD) were 95.6%, 69.2%, 47.6%, 53.3%, 61.5%, 67.3% and 49.9% respectively. Normal delivery (AOR 6.1, 95% CI 1.2–31.3) and higher maternal autonomy (AOR 5.2, 95% CI 1.8–14.6) were significantly associated with exclusive breastfeeding. Similarly, crop production and food security (AOR 3.8, 95% CI 1.9–7.7), maternal knowledge on MAD (AOR 2.5, 95% CI 1.0–6.2) and maternal autonomy (AOR 4.2, 95% CI 2.1–8.4) were significantly associated with minimum acceptable diet. Factors such as maternal education, maternal health services utilization, maternal knowledge, and maternal autonomy were associated with infant and young child feeding practices, which warrants further attention to these factors to reduce malnutrition.

**Funding:** NA received Dr. Harka Gurung - New ERA Fellowship 2019/20 from New ERA, Nepal. The funders had no role in study design, data collection and analysis, decision to publish, or preparation of the manuscript.

**Competing interests:** The authors have declared that no competing interests exist.

# Introduction

Infant and young child feeding (IYCF) is a key area to improve child survival and promote healthy growth and development [1]. The first two years of the child's life provide a critical window of opportunity to ensure survival, growth, and development through optimum infant and young child feeding practices [2]. Appropriate infant and young child feeding practices help to prevent almost 19% of all under-five deaths [3]. World Health Organization has recommended the initiation of breastfeeding within one hour of birth, exclusive breastfeeding for the first six months, and introduction of nutritionally adequate and safe complementary food at 6 months together with continuing breastfeeding up to two years of age or beyond [4].

Globally only about 44% of infants less than six months old were exclusively breastfed and 45% of the global child deaths are associated with under-nutrition. More than two-thirds of malnutrition-related child deaths are associated with inappropriate feeding practices during the first two years of life [5]. Improvement in infant and young child feeding practices is likely to reduce the burden of diarrhea-related morbidity and mortality [6].

The Nepal Demographic and Health Survey (NDHS) in 2016 reported about 66% of 0–5 months children were exclusively breastfeed in Nepal. Similarly, the prevalence of minimum meal frequency, minimum dietary diversity, and minimum acceptable diet was 71%, 47%, and 36% respectively. Breastfeeding and complementary feeding practices vary dramatically across different province of Nepal. More than fifty percent of children received a minimum acceptable diet in Gandaki province of Nepal, while this figure is only eighteen percentage in Province two of Nepal [7].

Many studies have been conducted to identify factors associated with infant feeding practices in Nepal. Evidence suggests that education, household wealth status, geographical location, maternal age, antenatal visit, post-natal visit, household food security, family community support are associated with exclusive breastfeeding and complementary feeding in Nepal [8–10]. Maternal factors such as decision-making capacity, education, knowledge, and maternal health services utilization are important factors for child feeding practices [10–13]. These factors pose better access and control over household resources to mothers which allows them to allocate resources to maintain and improve feeding practices [14].

The Government of Nepal has developed and implemented different acts, policies, strategies, and programs to improve infant feeding practices [15]. IYCF is a priority strategy of the Ministry of Health and Population. Despite the scale-up of this program was scaled up to all districts of Nepal, IYCF practices remain poor [16]. Therefore, it is more important to understand local and regional child feeding practices and their associated factors before developing strategies aimed to improve child feeding practices in Nepal. So, this study helps to determine factors associated with child feeding practices in the hilly region of Nepal and to generate local evidence for informed planning of the interventions to address poor IYCF practices.

# Materials and methods

## Study design and population

This was a community-based cross-sectional study conducted during September 2019 to August 2020 in Syangja district Nepal. Syangja is a part of Gandaki Province and is one of the 77 districts of Nepal. It covers an area of 1,164 km$^2$ (449 square miles). As per census 2011, The district has a population of 289,148 [17]. It is located 215 kilometers west of the capital city of Nepal. There are 20 hilly districts in Nepal where 45 percent of the total Nepalese population resides. Altogether there are five urban municipalities and six rural municipalities in Syangja district. The study area was two urban municipalities (Bhirkot and Chapakot) and two

rural municipalities (Phedikhola and Biruwa) of Syangja district. A ward is the smallest administrative unit in Nepal and the number of wards in a municipality ranges from a minimum of five to a maximum of 33. The study population was mothers and their children aged 0–23 months. Total population of child aged 0–23 months in Syangja district is 24,123 [17]. Considering the prevalence of minimum acceptable diet as 36%; at a 5% margin of error and 5% level of significance the final sample size was 360. The response rate in this study was 100%.

Multistage random sampling technique was used to select the participants. In the first stage, two rural municipalities and two urban municipalities were chosen randomly from the total list of municipalities. In the second stage, two wards from each municipality were selected by simple random sampling. In the final stage, participants were selected proportionate to population size from selected wards by using the list of under two-year children as a sampling unit. The list of the children aged 0–23 months was obtained from a comprehensive list maintained by female community health volunteers and immunization register maintained by health post.

## Data collection tools and variables

**Tools.** Face to face interview was conducted with 360 mothers having children less than 2 years in Syangja district. NA and five trained enumerators were involved in data collection from 5[th] January 2020 to 26[th] January 2020. The average time for an interview was 45 minutes.

A structured questionnaire was prepared. The study tool was adopted from Nepal Demographic Health Survey (NDHS) 2016 with necessary modification. The questionnaire was divided into six sections: socio-demographic and socio-economic information, maternal health services related information, infant and young child feeding practices related information, maternal knowledge about IYCF, information on agriculture-related practices, and information on maternal autonomy. The questionnaire included 69 questions.

**Outcome variables.** To assess child feeding practices, we used six core indicators of IYCF practices. These indicators were recorded using information about foods given to the child in the last 24 hours before the interview according to the definition of IYCF core indicators guidelines. According to WHO, IYCF indicators were defined as follows.

- Early initiation of breastfeeding was defined as the proportion of children aged 0–23 months who commenced breastfeeding within the first hour of birth.

- Exclusive breastfeeding was defined as the proportion of infants 0–5 months of age who were fed no other food or drink, not even water, except breast milk (including milk expressed or from a wet nurse), but allows the infant to receive oral rehydration salt, drops, and syrups (vitamins, minerals, and medicines).

- Introduction of complementary foods (solid, semi-solid, or soft foods) was defined as the proportion of infants who initiated solid, semi-solid, or soft foods in the six months of age.

- Minimum dietary diversity was defined as the proportion of children aged 6–23 months who received foods from four or more out of seven food groups. The seven food groups are grains, roots, and tubers; legumes and nuts; dairy products (milk, yogurt, cheese); flesh foods (meat, fish, poultry, and liver/organ meats); eggs; vitamin A-rich fruits and vegetables; other fruits and vegetables.

- Minimum meal frequency was defined as the proportion of breastfed and non-breastfed children 6–23 months of age who received solid, semi-solid, or soft food (including milk feed for non-breastfeed children) minimum number of times or more. The minimum number of times was defined as two times for breastfed infants 6–8 months; three times for breastfed children 9–23 months; four times for non-breastfed children 6–23 months.

- Minimum acceptable diet (MAD) was defined as the proportion of children 6–23 months of age who received a minimum acceptable diet (apart from breast milk). MAD is the sum of two fractions: (1) the proportion of breastfed children 6–23 months of age who had at least the minimum dietary diversity and the minimum meal frequency during the previous day; plus (2) the proportion of non-breastfed children 6–23 months of age who received at least two milk feedings and had at least the minimum dietary diversity and the minimum meal frequency during the previous day.

**Independent variables.** Independent variables were selected based on previously published studies. Independent variables were broadly classified into socio-demographic and economic-related factors, maternal health services-related factors, maternal knowledge-related factors, maternal autonomy-related factors, and agriculture-related factors.

Socio-demographic and economic factors included the age of the mother, age of the child, sex of child, religion, ethnicity, mother's occupation, mother's education, types of family, number of children, and wealth index [10]. The wealth index is a composite index calculated by using household assets. It was constructed by principal component analysis and categorized into first, 1; second,2; middle,3; fourth,4; and highest,5 [18]. Maternal health services-related information included the number of antenatal visits, postnatal visits, delivery place, types of delivery, growth monitoring, and nutritional counseling. Maternal knowledge-related variables included maternal knowledge on recommended feeding practices (first food, time of initiation of breast milk, frequency, variety, time to start semi-solid food, continuous breastfeeding). Agriculture-related variables included the production of crops, months enough for consumption, and home gardens [9,10]. Maternal autonomy included the decision-making power of mothers on the household decision (large household purchase and daily household needs), financial decision, child feeding, health care decision, and decision on mobility [19]. We constructed a nine-item questionnaire to assess the mother's autonomy. Four of the nine questions were adopted from Nepal Demographic and Health Survey questionnaire. Five other questions were constructed based on the review of previous studies. The final questionnaire measured different dimensions of a mother's autonomy including financial, child feeding, household related, healthcare-related as well as freedom of movement. Mothers were able to choose between five options ranging from 5 = decide by own self to 1 = decide by others. A continuous score was created by summing up the mother's responses to individual questions. The total score was further categorized into tertiles (high, medium, and low). From pretesting, Cronbach's alpha value of 0.84 was obtained for the mother autonomy scale, which indicated good internal consistency of the scale.

## Statistical analysis

Data were cleaned, coded, and entered in EpiData. The entered data was exported to Statistical Package for Social Sciences (SPSS) version 21 for analysis. Data analysis was done in three stages. In the first stage, descriptive analyses (frequency and percentage) were used to report the dependent and independent variables. Frequency tables were used for categorical variables, while mean and standard deviation (SD) were calculated for continuous variables.

The second stage of analysis involved testing for the association of independent variables with infant and young child feeding practices. Chi-square statistics and p-value at a 95% level of confidence were reported. The third stage involved testing the strength of association between dependent and independent variables using binary logistic regression (Table in S3 File). All variables significant at 5% significance level in the Chi-square test were subjected to multivariate logistic regression analysis. Independent variables with a P-value <0.05 were

entered into the multivariate analysis. The adjusted and unadjusted odds ratio with 95% confidence intervals were reported.

### Data quality control

The choice of study design and methods suited for this study was decided based on the review of scientific literature. To ensure content validity, the questionnaire was developed based on study objectives and variables. To ensure tool validity, the standard questionnaire was used with necessary modifications. For respondent validity, face to face interview with the mother was conducted for data collection. Data collection was done by the researcher himself for less interpersonal variation and completeness of data. The tool was pretested in ward number 1 of Chapakot municipality, which was not selected for the study.

### Ethical consideration

Approval was taken from the Institutional Review Committee of the Institute of Medicine and Central Department of Public Health, Tribhuvan University. Formal permission was also taken from the office of the respective urban and rural municipalities. The purpose and benefit of the study were clearly explained to the respondents and signed informed consent was obtained from the study participants before collecting data. Only participants who gave written or verbal consent took part in the study. Participation was strictly voluntary, and participants were free at any point in time to stop participation. Those participants who did not have adequate infant feeding practices were provided with the correct information and were advised to consult with health workers at any point of service outlets during data collection.

## Results

### General characteristics of study participants

Table 1 describes the socio-demographic characteristics of the study population. Out of 360 children, more than half (51.4%) were males. The age of study participants was between 15 to 40 years with a majority (40%) from the age group 25 to 29 years. Most children (71.4%) were from the age group 6 to 23 months. The mean age and SD of participants was 26.2 ± 4.1 years. More than half (54.7%) of participants were Brahmin/Chhettri and a majority (92.8%) of participants were Hindu. The majority (70%) of participants produced crops and was enough for 12 months.

### Distribution of participants by utilization of maternal health service-related information

Table 2 describes maternal health services utilization by mothers. Nearly half (49.7%) of participants took antenatal care services four or more times as per guidelines. Only 37.5% of participants went for post-natal services two or more times. A majority (83.3%) of mothers had delivered at the health facility. Three out of four (70.3%) participants took growth monitoring and promotion services and more than two out of five (44%) of participants received nutritional counseling from health workers.

### Distribution of participants by their child feeding practice

Table 3 describes child feeding practices. A majority (95.6%) of participants fed breast milk to their children. Among them, more than two out of three (69.2%) of participants initiated breastfeeding immediately within an hour of childbirth. Less than half (47.6%) of children aged 0–5 months were exclusively breastfed. More than half (53.3%) of children aged 6–23

**Table 1. Distribution of participants by socio-demographic characteristics.**

| Characteristics | Number | Percentage |
|---|---|---|
| **Sex of child** | | |
| Female | 175 | 48.6 |
| Male | 185 | 51.4 |
| **Mother age (years)** | | |
| 15–24 | 138 | 38.3 |
| 25–34 | 211 | 58.6 |
| 35 and above | 11 | 3.1 |
| Mean ± SD | 26.2 ± 4.1 | |
| **Children age (months)** | | |
| 0–5 | 103 | 28.6 |
| 6–23 | 257 | 71.4 |
| Mean ± SD | 10 ± 6.1 | |
| **Ethnicity** | | |
| Brahmin/chhetri | 197 | 54.7 |
| Janajati | 99 | 27.5 |
| Dalit | 62 | 17.2 |
| Thakuri | 2 | 0.2 |
| **Family type** | | |
| Nuclear | 159 | 44.2 |
| Joint | 201 | 55.8 |
| **Religion** | | |
| Hindu | 334 | 92.8 |
| Buddhist | 23 | 6.4 |
| Christian | 3 | .8 |
| **Education** | | |
| Illiterate | 16 | 4.4 |
| Informal or primary | 76 | 21.1 |
| Secondary or above | 268 | 74.4 |
| **Occupation** | | |
| Agriculture | 193 | 53.6 |
| Housewife | 69 | 19.2 |
| Service | 65 | 18 |
| Business | 33 | 9.2 |
| **Number of children** | | |
| 1 | 133 | 36.9 |
| 2 | 160 | 44.4 |
| 3 | 59 | 16.4 |
| 4 or more | 8 | 2.2 |
| **Crop production and food security** | | |
| No crop produced or not enough food for 12 months | 108 | 30 |
| Crop produced and enough food for 12 months | 252 | 70 |

months were initiated with complementary feeding timely. Similarly, nearly half (49.4%) of children aged 6–23 months were fed according to recommended IYCF practice.

The bivariate analysis has been reported separately as a supplementary file (Table in S3 File).

**Table 2. Distribution of participants by utilization of maternal health service.**

| Maternal health services | Number | Percentage |
|---|---|---|
| **ANC visit** | | |
| None | 29 | 8.1 |
| 1 to 3 times | 152 | 42.2 |
| 4 or more time | 179 | 49.7 |
| **PNC visit** | | |
| None | 80 | 22.2 |
| 1 time | 145 | 40.3 |
| 2 or more time | 135 | 37.5 |
| **Place of delivery** | | |
| Institution | 300 | 83.3 |
| Home | 60 | 16.7 |
| **Types of delivery** | | |
| Normal vaginal | 322 | 89.4 |
| Cesarean Section | 38 | 10.6 |
| **Growth monitoring** | | |
| Yes | 253 | 70.3 |
| No | 107 | 29.7 |
| **Received nutrition counseling** | | |
| Yes | 161 | 44.7 |
| No | 199 | 55.3 |

## Factors associated with timely initiation of breastfeeding

Table 4 shows the factors associated with the timely initiation of breastfeeding. The result from the regression analysis showed that timely initiation of breastfeeding was more likely among mothers who delivered normally (AOR 10.7, 95% CI 4.5–25.3), belonged to the nuclear family (AOR 2.1,95% CI 1.2–3.5), and had correct knowledge on time of initiation of breastfeeding (AOR 9.1, 95% CI 3.5–23.6).

## Factors associated with exclusive breastfeeding

Table 5 shows the factors associated with exclusive breastfeeding. The result from the regression analysis showed that exclusive breastfeeding was more likely among mothers who delivered through normal vaginal delivery (AOR 6.1, 95% CI 1.2–31.3) and mothers with the highest (AOR 5.2, CI 1.8–14.6) and middle (AOR 3.1, CI 1.0–9.2) level of autonomy.

## Factors associated with timely initiation of complementary feeding

Table 6 shows the factors associated with the timely initiation of complementary feeding. The result from regression analysis showed that timely initiation of complementary feeding was more likely for a male child (AOR 11.7, 95% CI 6.0–22.6) and mothers who received nutritional counseling (AOR 2.7, 95% CI 1.2–6.0). The result also shows that timely initiation of complementary feeding was less likely among mothers who had more children in their family (AOR 0.2, 95% CI 0.1–0.6).

## Factors associated with minimum acceptable diet (MAD)

Table 7 shows the factors associated with a minimum acceptable diet. The result from the regression analysis showed that minimum acceptable diet was more likely for mother who

**Table 3. Distribution of participants by their child feeding practices.**

| Feeding practices | Number | Percentage |
|---|---|---|
| **Breastfeeding** | | |
| Yes | 344 | 95.6 |
| No | 16 | 4.4 |
| **Initiation of breastfeeding** | | |
| Immediately within one hour | 238 | 69.2 |
| Within 1 to 24 hours | 80 | 23.3 |
| After 1 day | 15 | 4.4 |
| Do not know | 11 | 3.2 |
| **Colostrum milk** | | |
| Feed | 312 | 90.7 |
| Not feed | 32 | 9.3 |
| **Breast milk substitute** | | |
| Infant formula | 11 | 68.8 |
| Other milk | 5 | 31.3 |
| **Exclusive breastfeeding** | | |
| Not exclusive breastfeed | 54 | 52.4 |
| Exclusive breastfeed | 49 | 47.6 |
| **Initiation of complementary feeding** | | |
| Not timely initiation | 120 | 46.7 |
| Timely initiation | 137 | 53.3 |
| **Minimum dietary diversity** | | |
| Not met | 99 | 38.5 |
| Met | 158 | 61.5 |
| **Minimum meal frequency** | | |
| Not met | 84 | 32.7 |
| Met | 173 | 67.3 |
| **Minimum acceptable diet** | | |
| Not met | 130 | 50.6 |
| Met | 127 | 49.4 |

**Table 4. Factors associated with timely initiation of breastfeeding.**

| Variables | Unadjusted OR | 95% CI | P-value | Adjusted OR | 95% CI | P-value |
|---|---|---|---|---|---|---|
| **Type of delivery** | | | | | | |
| Cesarean Section | Ref. | | | Ref. | | |
| Normal | 9.3 | 4.0–21.4 | <0.001* | 10.7 | 4.5–25.3 | <0.001* |
| **Type of family** | | | | | | |
| Joint | Ref. | | | Ref. | | |
| Nuclear | 1.8 | 1.1–2.9 | 0.012* | 2.1 | 1.2–3.5 | 0.006* |
| **Knowledge on initiation of breastfeeding** | | | | | | |
| Incorrect | Ref. | | | Ref. | | |
| Correct | 7.2 | 2.9–17.7 | <0.001* | 9.1 | 3.5–23.6 | <0.001 |

Note

* statistically significant at 95% level of confidence.

**Table 5. Factors associated with exclusive breastfeeding.**

| Variables | Unadjusted OR | 95% CI | P-value | Adjusted OR | 95% CI | P-value |
|---|---|---|---|---|---|---|
| **Delivery type** | | | | | | |
| CS | Ref. | | | Ref. | | |
| Normal | 1.5 | 0.9–22.9 | 0.050 | 6.1 | 1.2–31.3 | 0.029* |
| **Autonomy** | | | | | | |
| Lowest | Ref. | | | Ref. | | |
| Middle | 2.6 | 0.9–7.4 | 0.063 | 3.1 | 1.0–9.2 | 0.033* |
| Highest | 4.5 | 1.6–12.5 | 0.003 | 5.2 | 1.8–14.6 | 0.002* |

Note

* statistically significant at 95% level of confidence.

produces crops and enough for 12 months (AOR 3.8, 95% CI 1.9–7.7), mothers who had correct knowledge on the minimum acceptable diet (AOR 2.5, 95% CI 1.0–6.2), mothers with middle (AOR 4.2, 95% CI 2.1–8.4) and higher (AOR 3.8, CI 1.8–7.7) level of autonomy.

## Discussion

Our study found that almost all children were breastfed, among them the majority of the children were initiate breastfeeding within one hour of delivery. The percentage of mothers who performed early initiation of breastfeeding in our study is comparable to a study conducted in central Nepal [20] but more than the national average of 54.9% reported by NDHS 2016, 48% reported by a recent national study [21], 57% reported in Bhaktapur [13] and 37.1% reported in Satar community Nepal [22]. Similarly, the rate is higher when compared with studies conducted in other South Asian countries like India (36.4%) [23], Bangladesh (24%) [24], Pakistan

**Table 6. Factors associated with timely initiation of complementary feeding.**

| Variables | Unadjusted OR | 95% CI | P-value | Adjusted OR | 95% CI | P- value |
|---|---|---|---|---|---|---|
| **Sex** | | | | | | |
| Female | Ref | | | Ref | | |
| Male | 8.0 | 4.6–14.1 | <0.001 | 11.7 | 6.0–22.6 | <0.001* |
| **Child number** | | | | | | |
| 1 | Ref | | | Ref | | |
| 2 | 1.0 | 0.5–1.7 | 0.965 | 1.4 | 0.7–2.7 | 0.305 |
| 3 | 0.2 | 0.1–0.5 | 0.001 | 0.2 | 0.1–0.6 | 0.006* |
| 4 or more | 0.8 | 0.1–3.5 | 0.796 | 0.7 | 0.1–3.5 | 0.681 |
| **Delivery place** | | | | | | |
| Home | Ref | | | Ref | | |
| Institution | 2.1 | 1.0–4.2 | 0.028 | 1.8 | 0.7–4.4 | 0.154 |
| **Growth monitoring** | | | | | | |
| No | Ref | | | Ref | | |
| Yes | 1.8 | 1.1–3.2 | 0.018 | 1.1 | 0.5–2.5 | 0.756 |
| **Nutrition counseling** | | | | | | |
| Not Received | Ref | | | Ref | | |
| Received | 1.9 | 1.1–3.2 | 0.011 | 2.7 | 1.2–6.0 | 0.015* |

Note

* statistically significant at 95% level of confidence.

**Table 7. Factors associated with minimum acceptable diet.**

| Variables | Unadjusted OR | 95% CI | P- value | Adjusted OR | 95% CI | p- value |
|---|---|---|---|---|---|---|
| **Crop production and food security** | | | | | | |
| No crop produced or food not enough for 12 months | Ref | | | Ref | | |
| Crop produced and food enough for 12 months | 4.3 | 2.2–8.3 | <0.001 | 3.8 | 1.9–7.7 | <0.001* |
| **Delivery place** | | | | | | |
| Home | Ref | | | Ref | | |
| Institution | 2.1 | 1.0–4.2 | 0.028 | 1.3 | 0.5–3.3 | 0.526 |
| **PNC visit** | | | | | | |
| None | Ref. | | | Ref. | | |
| 1 time | 2.1 | 1.0–4.2 | 0.030 | 1.2 | 0.5–3.0 | 0.587 |
| 2 or more times | 2.3 | 1.1–4.6 | 0.016 | 1.3 | 0.5–3.3 | 0.562 |
| **Nutrition counseling** | | | | | | |
| No | Ref. | | | Ref. | | |
| Yes | 2.0 | 1.2–3.4 | 0.005 | 1.3 | 0.7–2.4 | 0.291 |
| **Knowledge on MAD** | | | | | | |
| Incorrect | Ref. | | | Ref. | | |
| Correct | 2.3 | 1.0–5.2 | 0.033 | 2.5 | 1.0–6.2 | 0.043* |
| **Autonomy** | | | | | | |
| Lowest | Ref. | | | Ref. | | |
| Middle | 4.6 | 2.4–8.8 | <0.001 | 4.2 | 2.1–8.4 | <0.001* |
| Highest | 3.5 | 1.8–6.8 | <0.001 | 3.8 | 1.8–7.7 | <0.001* |

Note

* statistically significant at 95% level of confidence.

(8.5%) [25], Ethiopia (57.4) [26]. This might be due to increasing access to maternal health services, effective intervention, and commitments to promote breastfeeding through nutrition programs in Nepal. Our study showed a positive association between type of delivery, type of family, and maternal knowledge on the correct time of initiation of breastfeeding with timely initiation of breastfeeding. Similar findings were reported by other studies conducted in Nepal [27,28]. Timely initiation of breastfeeding was significantly higher among mothers who had a normal vaginal delivery. This finding was in agreement with the study conducted in India [29]. Longer procedure, pain following the procedure, severe bleeding, effects of anesthesia, and tiredness associated with caesarean delivery make it difficult to initiate breastfeeding early [30]. Similar to other studies from Bangladesh [24], Nepal [28], China [31], our study also found a positive association between timely initiation of breastfeeding and mother's knowledge. Mother's knowledge helps to adopt positive feeding practices. Mothers who were from nuclear family were more likely to initiate breastfeeding on time. A similar finding was reported from a study in Cartagena city of Colombia [32]. This might be due to children who live with nuclear family have high support for the mother to initiate breastfeeding. Focuses on improving delivery assistance at health institution, promotion of kangaroo mother care practices soon after delivery, provide counseling during antenatal care visit and improve maternal health care service utilizations would facilitate the timely initiation of breastfeeding.

In our study, the prevalence of exclusive breastfeeding was 47.6%, which was higher than the global average rate of 40% [33] and lower than the national average rate of 66% [34]. This finding was higher than the study conducted in the mid-western and eastern regions of Nepal [35] and slightly lower than the study conducted in rural southern Nepal [11]. This variation might be due small sample size compared to this study. We found mothers who delivered by

cesarean section were less likely to exclusively breastfeed. This finding was consistent with the study conducted in India [36]. Our study also found that mothers with highest autonomy were more likely to exclusively breastfeed. This finding is similar to the studies conducted in Saharan Africa, south Asia and Latin America [37] in Vietnam [38]. Mothers with higher autonomy have higher decision-making capacity and they can decide themselves on matters related to child feeding, access to resource, utilization of health services, access to information and financial protection which ultimately improve the feeding practices.

In our study timely initiation of complementary feeding was 53.3% which was lower than the recent NDHS 2016 survey and similar to the other study conducted in other parts of the country [10,39]. Studies conducted in other developing countries like Bangladesh [40] and Ethiopia [41] also reported higher percentages (>60%) of complementary feeding at six months. Mothers who had more children were less likely to initiate complementary feeding at six months of age which was similar to findings from Hongkong [42] and Finland [43]. Mothers who received nutritional counseling were more likely to initiate complementary feeding on time. A similar finding was reported from a study in the Satar community [22]. Counseling mothers about complementary feeding during post-natal and growth monitoring phases has a favorable impact on the timely initiation of complementary feeding. Similarly, timely initiation of complementary feeding was significantly associated with the sex of children. Male children were more likely to have had timely initiation of complementary feeding. A similar finding was reported from a study conducted in other parts of Nepal [44] which reflects the long-standing traditional gender norm that discriminates against timely feeding of a female child [45].

In our study minimum dietary diversity, minimum meal frequency and minimum acceptable diet were 61.5%, 67.3%, and 49.4% respectively. These percentages are lower than the recent Nepal Demographic and Health Survey report. The percentage of minimum meal frequency was lower than the study conducted in the Terai region of Nepal which reported 84% of children meet minimum meal frequency [10]. This may be due to better access to maternal health services. A higher percentage of children met minimum dietary diversity in Sri Lanka 71% [26] and Bangladesh 81.1% [26] which may be because the majority of rural Nepalese communities depend on specific staple food available locally such as rice, wheat, and potato. Our study found that mothers who produced crops enough for 12 months were more likely to meet the minimum acceptable diet. A similar finding was reported by the further analysis of NDHS study 2016 [16]. Mothers who had correct knowledge on recommended feeding practices were more likely to meet the minimum acceptable diet [46]. Mothers with higher decision-making capacity were more likely to meet the minimum acceptable diet [47]. In our study mothers having middle autonomy had the best complementary feeding practices. A potential explanation might be that while low autonomy reduces women's access to and control over resources in the household, the highest scores in autonomy might imply a lower level of partner support and thus more responsibility for women, which could reduce their caregiving capacity. Furthermore, women in the middle autonomy may be deciding jointly with their partners. This might reflect good relations and better communication between partners which might have resulted in better child complementary feeding practices. Others factors like antenatal visits, post-natal visits, wealth quintile, and the number of children are important determinants of child feeding practices in Nepal [20,27]. In our study these factors were not statistically associated due to variation in population size, setting, existing local traditions and beliefs. So further research with higher sample size and robust design is needed.

Some limitations need to be considered when interpreting the results of this study. WHO developed eight core indicators for assessing infant and young child feeding practices. In this study, only six indicators were used to assess the feeding practices of children. Two indicators

i.e. consumption of iron rich or iron fortified food and continue breastfeeding for one-year age are not included in this study. Our study had less sample size for effective exclusive breast-feeding practices. Another limitation of this study was recall bias due to the retrospective nature of the data collection, possibly resulting in over or underestimation of actual feeding practices. Although recall biases cannot be avoided, the researcher conducted all interviews by asking probing questions to gather exact information. It should be also noted that epidemiological studies of this kind do not establish causality but may suggest associations. The use of already validated NDHS tools, WHO guidelines, indicators, and appropriate statistical adjustment are strengths of this study.

## Conclusion

Our study presents important findings on infant and young child feeding practices and their associated maternal factors among mothers of under two years' children in Syangja district of Nepal. Early initiation of breastfeeding was good but feeding according to recommended IYCF practice was poor (i.e., less than half). It is evident from this study that the factors such as maternal education, nutritional counseling, food security, child numbers, maternal health services, decision-making power were identified as key factors associated with feeding practices and these factors should be carefully considered when designing strategies and interventions.

## Supporting information

**S1 File. Information sheet and consent form.**
(DOCX)

**S2 File. Conceptual framework.**
(DOCX)

**S3 File. Bivariate analysis.**
(DOCX)

**S1 Questionnaire.**
(DOCX)

**S1 Data.**
(SAV)

## Acknowledgments

We would like to express our deep sense of gratitude to Dr. Amod Kumar Poudyal, Mr. Rajan Paudel, and Dr. Khem Karki for their valuable support and suggestions. We acknowledge the support from the Health Coordinator of Phedikhola rural municipality, Bhirkot municipality, Chapakot municipality, and Biruwa rural municipality for coordination at study sites. We also acknowledge the support from Mr. Ashish Timalsina, Mr. Rabindra Bhandari, Ms. Jijeebisha Baral, Ms. Pratibha Thapa, and Ms. Rama Bhandari.

## Author Contributions

**Conceptualization:** Nabin Adhikari, Dipak Prasad Upadhya, Sumita Pathak, Pranil Man Singh Pradhan.

**Data curation:** Nabin Adhikari, Sachin Pokharel.

**Formal analysis:** Nabin Adhikari, Sachin Pokharel.

**Funding acquisition:** Kiran Acharya.

**Methodology:** Nabin Adhikari, Kiran Acharya, Dipak Prasad Upadhya, Pranil Man Singh Pradhan.

**Resources:** Nabin Adhikari, Dipak Prasad Upadhya.

**Software:** Kiran Acharya, Dipak Prasad Upadhya.

**Supervision:** Nabin Adhikari, Kiran Acharya, Pranil Man Singh Pradhan.

**Validation:** Nabin Adhikari, Dipak Prasad Upadhya, Pranil Man Singh Pradhan.

**Writing – original draft:** Nabin Adhikari.

**Writing – review & editing:** Kiran Acharya, Sumita Pathak, Pranil Man Singh Pradhan.

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
