## [Decision Letter · Decision Letter 0]

23 Feb 2021

PONE-D-21-02451

Infant and young child feeding practices and its association with maternal factors among mothers of under two years children in western hilly region of Nepal

PLOS ONE

Dear Dr.Pranil Man Singh Pradhan,

Thank you for submitting your manuscript to PLOS ONE. After careful consideration, we feel that it has merit but does not fully meet PLOS ONE’s publication criteria as it currently stands. Therefore, we invite you to submit a revised version of the manuscript that addresses the points raised during the review process.

We look forward to receiving your revised manuscript.

Kind regards,

Claudia Marotta

Academic Editor

PLOS ONE

4. We noticed you have some minor occurrence of overlapping text with the following previous publications, which needs to be addressed:

- "Associations between infant and young child feeding practices and diarrhoea in Indian children: a regional analysis" by Dhami et al., preprint posted 05 Feb 2020 (https://www.researchsquare.com/article/rs-13231/v1).

- "Addressing barriers to exclusive breast-feeding in low- and middle-income countries: a systematic review and programmatic implications" by Kavle et al., Public Health Nutrition Vol. 20(17), 2017 (https://www.cambridge.org/core/journals/public-health-nutrition/article/addressing-barriers-to-exclusive-breastfeeding-in-low-and-middleincome-countries-a-systematic-review-and-programmatic-implications/53EBA65F5D58D16E3E4D32E0FCFA938B).

- "Dietary diversity, meal frequency and associated factors among infant and young children in Northwest Ethiopia: a cross- sectional study" by Beyene et al., BMC Public Health 15, 2015 (https://bmcpublichealth.biomedcentral.com/articles/10.1186/s12889-015-2333-x).

In your revision ensure you cite all your sources (including your own works), and quote or rephrase any duplicated text outside the methods section. Further consideration is dependent on these concerns being addressed.

Additional Editor Comments:

dear authors follow reviewer suggestion to improve your paper

Reviewers' comments:

Reviewer's Responses to Questions

**Comments to the Author**

1. Is the manuscript technically sound, and do the data support the conclusions?

Reviewer #1: Yes

Reviewer #2: No

2. Has the statistical analysis been performed appropriately and rigorously? 

Reviewer #1: Yes

Reviewer #2: No

3. Have the authors made all data underlying the findings in their manuscript fully available?

Reviewer #1: Yes

Reviewer #2: Yes

4. Is the manuscript presented in an intelligible fashion and written in standard English?

Reviewer #1: Yes

Reviewer #2: Yes

5. Review Comments to the Author

Reviewer #1: Congratulations on your article! It is notable that it required hard working.

Methods are detailed and make readers understand your trains of thought.

Results are clear, although it would be nice to show data about exclusive breastfeeding in children from 6-23 months. It would be interesting to check if the children exclusively breastfed at age 0-5 months had better feeding practices than children not breastfed. Besides, add information about how long these children were breastfeed would clarify feeding habits of the studied population and all social aspects involved.

Data shown on Table 3 relate to exclusive breastfeeding of the 103 children from 0-5 months. Data about the other 237 children would increase the interest on your article and would add relevant information to your research.

Discussion is well written and full of valid references. Comparing your data to data from countries with similar social issues enable readers to understand the range of your article.

Reviewer #2: Review

I am glad for the opportunity to review this manuscript. IYCF practice is very important issue for the context of most low and middle income countries. The main objective of this article is to associate the IYCF practices with maternal health care seeking practices. However, the plausible hypothetical association of maternal health care seeking practices with some of the distal IYCF practices is not well developed in the child nutrition literature.

According to my observations, there are several scientific drawbacks of this manuscript, especially in the conceptual framework formulation and Statistical analysis. Please see my detailed comments below.

Title page:

Line 5-24: The authorship order is bit confusing. The lead authorship, corresponding author and senior authorship is not clear. It would be better if the lead or senior author become the corresponding author. Moreover, the equal contribution in the middle is not clear to me. Equal contribution is mostly applicable for lead author and senior author.

Introduction

1. Line 54-55: When the complementary feeding should be started?

2. Line 61-66: Now it is 2021. Is there any updated data available? Please use the updated evidence throughout the Introduction section.

3. Line 61-72: Concise the write-up. The concept is repeating

4. Line 73-82: Why the autonomy was repeated and emphasized so much? But nothing told about the maternal care seeking practice indicators.

5. Line 73-85: The problem identification of necessities to explore the association maternal health seeking practice is not well defined. That means the rationale of the objective is not enough convincing.

Here the authors need to mention the available determinates found in the available similar literatures of Nepal and similar countries. Authors also need to find out at least one paper that explored the association of the maternal care seeking factors and child IYCF practices. If no specific paper found then the biological plausibility need to establish by the convincing argument. Hence, the conceptual framework needs to develop with the plausible independent factors. MNH practices could be one of the domains of conceptual framework and others factors will be the covariates/confounders.

Materials and methods

1. Line 98-99: Tell how many of districts 77 district are similar to Syangja and what is the total population of those similar district? Otherwise your sample only represents the limited 289,148 population of one district. Policy makers may not have the importance to the limited number of population. If possible then add a map.

2. Line 99: 1.164 km2 must be corrected.

3. Line 104-106: From how many rural and how many urban municipalities? Is rural administrative unit also called municipality in Nepal? What is the range of number of wards in municipalities?

4. Line 107-108: Not clear. What is proportionate to population size? Did authors make a list by themselves and randomly select the participants or something else? Please write what actually done there? Sometime the extensive sampling process may not possible to follow for the time and cost constraint. This could be a possible limitation of study.

5. Line 109-110: Add reference of minimum acceptable diet as 36% and total under two years children in Syangja district as 10524.

6. Line132-155: Please reduce the text. Only MAD was analyzed as main outcome variable. It is better to reduce text throughout outcome variable definitions.

7. Line 157-160: Need to add citation for all the independent variables. All the factors are the covariates/confounders between maternal health service factors and IYCF. It is better to move this part in Introduction and aligned with the conceptual framework formulation.

8. Line 174-176: How many researchers collected data? It is very difficult to conduct 360 interviews in 21 days by 1 person. What is the average time of interview?

9. Line 173-189: Move the data collection tools section before the variable section. If possible then aligned the text of both section and make one broad section named “Data collection tools and variables”. Firstly, write about tools. Then under this broad section, write outcome variables’ and independent variables’ sub section.

10. Line 206-209: VIF>10 is mostly useful for continuous independent variables. Please revisit other standard way for detecting multi-colinearity for categorical variables.

11. Line 219-220: Which indicators or scale was used to calculate this Cronbach’s alpha?

Results

1. Line 198-206 and line 258-291: I have some major observations in the data analysis. I think data analysis did not conducted properly as written in Statistical analysis section.

Here there are four main outcome variables and many independent variables. Firstly, chi-square test and bi-variate logistic regression need to do for each outcome variable with all relevant independent variables. Then all significant independent variables found in chi-square test or bi-variate logistic analysis need to insert in multivariate regression. You may consider the multi-collinearly or overlapping nature of independent variables. For all 4 outcomes, you have to repeat this analysis. The final interpretation should be based on adjusted analysis. Please see some format of other published articles. If you don’t want to put the elaborate analysis in the main manuscript then you can only put the result of adjusted analysis in the manuscript. But you have to put the elaborate analysis in a supplementary file.

My main concern is, some most important individual or household level determinants (e.g., mother age, education, employment, number of child/parity, wealth quintile, ethnicity, family type, food insecurity status, child age, sex etc) may not included in the bi-variate and multivariate analysis. For this reason, the effect sizes between bvariate and multivariate regression did not changed more. Please revise the full analysis.

2. Table 7: The crude OR and Adjusted OR for ANC and PNC reversed completely. There might be some problem in somewhere. Please consult with a public health statistical expert.

Discussion and conclusion

1. Based on the available result, the discussion and conclusion was well written. But full discussion and a conclusion may need to revise based on revised result.

2. My concluding remarks and suggestion to the authors is that if most of the maternal health utilization factors would not significant in the revised adjusted data analysis, then to change objective and title as general determinant exploration of IYCF practices.

6. PLOS authors have the option to publish the peer review history of their article (what does this mean?). If published, this will include your full peer review and any attached files.

Reviewer #1: No

Reviewer #2: **Yes: **Md. Rashidul Azad

Research Investigator,

Maternal and Child Health Division,

icddr,b

---

## [Author Response · Author response to Decision Letter 0]

29 Mar 2021

Reviewer #1: Congratulations on your article! It is notable that it required hard working.

Methods are detailed and make readers understand your trains of thought.

Results are clear, although it would be nice to show data about exclusive breastfeeding in children from 6-23 months. It would be interesting to check if the children exclusively breastfed at age 0-5 months had better feeding practices than children not breastfed. Besides, add information about how long these children were breastfeed would clarify feeding habits of the studied population and all social aspects involved. Data shown on Table 3 relate to exclusive breastfeeding of the 103 children from 0-5 months. Data about the other 237 children would increase the interest on your article and would add relevant information to your research.

Response- Thank you for your comments. As per assessment tools for infant and young child feeding practices we can assess the exclusive breastfeeding of 0-5 months children only so this tools is not valid to assess exclusive breastfeeding in children from 6-23 months. 

Discussion is well written and full of valid references. Comparing your data to data from countries with similar social issues enable readers to understand the range of your article.

Response- Thank you for the comment. We have added relevant references. 

Reviewer #2: Review

Title page:

Line 5-24: The authorship order is bit confusing. The lead authorship, corresponding author and senior authorship is not clear. It would be better if the lead or senior author become the corresponding author. Moreover, the equal contribution in the middle is not clear to me. Equal contribution is mostly applicable for lead author and senior author.

Response- Thank you for the comment. We have amended the order of authorship.

Introduction

1. Line 54-55: When the complementary feeding should be started?

Response- Complementary feeding should be started after six months of age. We have mentioned it now on final version of manuscript.

2. Line 61-66: Now it is 2021. Is there any updated data available? Please use the updated evidence throughout the Introduction section.

Response- Thank you for the comment. We have now updated the data. 

3. Line 61-72: Concise the write-up. The concept is repeating

Response- Thank you for the comment. We have shortened it.

4. Line 73-82: Why the autonomy was repeated and emphasized so much? But nothing told about the maternal care seeking practice indicators.

Response-Thank you for the comment. We have now included maternal care seeking related evidence from Nepal. 

5. Line 73-85: The problem identification of necessities to explore the association maternal health seeking practice is not well defined. That means the rationale of the objective is not enough convincing.

Response- Thank you for the comment. We have explored the evidence related to maternal health services and infant and young child feeding practices. 

Here the authors need to mention the available determinates found in the available similar literatures of Nepal and similar countries. Authors also need to find out at least one paper that explored the association of the maternal care seeking factors and child IYCF practices. If no specific paper found then the biological plausibility need to establish by the convincing argument. Hence, the conceptual framework needs to develop with the plausible independent factors. MNH practices could be one of the domains of conceptual framework and others factors will be the covariates/confounders.

Response- Thank you for the comment. We have explored the evidence related to maternal health services and infant and young child feeding practices and provided two references (Reference 6 and 7) for the same. Conceptual framework attached as supplementary file S5.

Materials and methods

1. Line 98-99: Tell how many of districts 77 district are similar to Syangja and what is the total population of those similar district? Otherwise your sample only represents the limited 289,148 population of one district. Policy makers may not have the importance to the limited number of population. If possible then add a map.

Response- Thank you for the comment. Syangja district is one of the 20 hilly districts of Nepal. Our country has been divided into three ecological zone (i.e. mountain, hill, and terai) based on geographical situation and seventy-seven district based on population size, geographical location. About 45 percent of total population of Nepal live in the hilly districts and we have mentioned this in the methods.

2. Line 99: 1.164 km2 must be corrected.

Response- Thank you. We have corrected it. 

3. Line 104-106: From how many rural and how many urban municipalities? Is rural administrative unit also called municipality in Nepal? What is the range of number of wards in municipalities?

Response- Thanks for the comment. Altogether there are five urban municipalities and six rural municipalities in Syangja district. Rural administrative unit also called as rural municipality in Nepal. A ward is the smallest administrative unit in Nepal and number of wards in a municipality ranges from minimum of five to maximum of 33. We have mentioned this is the methods.

4. Line 107-108: Not clear. What is proportionate to population size? Did authors make a list by themselves and randomly select the participants or something else? Please write what actually done there? Sometime the extensive sampling process may not possible to follow for the time and cost constraint. This could be a possible limitation of study.

Response- Thank you for the comment. Proportionate to population size is method of sampling in which required number of samples was identified based on proportion of sample where the probability of selecting a unit is proportional to its size. Authors had access to list of under two years children from health post of the ward and records from female community health volunteers. Author used this list as a sample frame. Regarding municipalities and ward, authors had access to list of municipalities and wards from Provincial Administrative Office. 

5. Line 109-110: Add reference of minimum acceptable diet as 36% and total under two years children in Syangja district as 10524.

Response- Thank you for the comment. Reference has been added.

6. Line132-155: Please reduce the text. Only MAD was analyzed as main outcome variable. It is better to reduce text throughout outcome variable definitions.

Response- Thank you for the comment. We have done it.

7. Line 157-160: Need to add citation for all the independent variables. All the factors are the covariates/confounders between maternal health service factors and IYCF. It is better to move this part in Introduction and aligned with the conceptual framework formulation.

Response- Thank you for the comment. We have done this.

8. Line 174-176: How many researchers collected data? It is very difficult to conduct 360 interviews in 21 days by 1 person. What is the average time of interview?

Response- Thank you for your comment. Data collection team included five trained enumerators and author himself. Average time to taken interview was 45 min. We have mentioned this in the tools section.

9. Line 173-189: Move the data collection tools section before the variable section. If possible then aligned the text of both section and make one broad section named “Data collection tools and variables”. Firstly, write about tools. Then under this broad section, write outcome variables’ and independent variables’ sub section.

Response- Thank you for the comment. We have done accordingly.

10. Line 206-209: VIF>10 is mostly useful for continuous independent variables. Please revisit other standard way for detecting multi-colinearity for categorical variables.

Response- Thank you for the comment. We used Chi square test for categorical variables, and all variables significant at 5% level of significance in bivariate analysis were considered for multivariate analysis.

11. Line 219-220: Which indicators or scale was used to calculate this Cronbach’s alpha?

Response- Thank you for the comment. To check internal consistency in the mother’s autonomy scale, Cronbach’s alpha was used. 

Results

1. Line 198-206 and line 258-291: I have some major observations in the data analysis. I think data analysis did not conducted properly as written in Statistical analysis section.

Here there are four main outcome variables and many independent variables. Firstly, chi-square test and bi-variate logistic regression need to do for each outcome variable with all relevant independent variables. Then all significant independent variables found in chi-square test or bi-variate logistic analysis need to insert in multivariate regression. You may consider the multi-collinearly or overlapping nature of independent variables. For all 4 outcomes, you have to repeat this analysis. The final interpretation should be based on adjusted analysis. Please see some format of other published articles. If you don’t want to put the elaborate analysis in the main manuscript then you can only put the result of adjusted analysis in the manuscript. But you have to put the elaborate analysis in a supplementary file. My main concern is, some most important individual or household level determinants (e.g., mother age, education, employment, number of child/parity, wealth quintile, ethnicity, family type, food insecurity status, child age, sex etc) may not included in the bi-variate and multivariate analysis. For this reason, the effect sizes between bvariate and multivariate regression did not changed more. Please revise the full analysis.

Response- Thank you for the comment. We have included bivariate analysis as supplementary file (S6). 

2. Table 7: The crude OR and Adjusted OR for ANC and PNC reversed completely. There might be some problem in somewhere. Please consult with a public health statistical expert.

Response- Thank you for the comment. We have revised the analysis. 

Discussion and conclusion

1. Based on the available result, the discussion and conclusion was well written. But full discussion and a conclusion may need to revise based on revised result.

Response- Thank you for the comment. Discussion and conclusion have been revised. 

2. My concluding remarks and suggestion to the authors is that if most of the maternal health utilization factors would not significant in the revised adjusted data analysis, then to change objective and title as general determinant exploration of IYCF practices.

Response- Thank you for the comment. Factors such as maternal education, maternal health service utilization, maternal knowledge, decision making power are still significantly associated with IYCF practices in revised data analysis.

---

## [Decision Letter · Decision Letter 1]

7 May 2021

PONE-D-21-02451R1

Infant and young child feeding practices and its association with maternal factors among mothers of under two years children in western hilly region of Nepal

PLOS ONE

Dear Dr. Pradhan,

Thank you for submitting your manuscript to PLOS ONE. After careful consideration, we feel that it has merit but does not fully meet PLOS ONE’s publication criteria as it currently stands. Therefore, we invite you to submit a revised version of the manuscript that addresses the points raised during the review process.

We look forward to receiving your revised manuscript.

Kind regards,

Lucinda Shen

Associate Editor

on behalf of 

Claudia Marotta 

Academic Editor 

PLOS ONE 

Journal Requirements:

Additional Editor Comments (if provided):

congratulations

Reviewers' comments:

Reviewer's Responses to Questions

**Comments to the Author**

1. If the authors have adequately addressed your comments raised in a previous round of review and you feel that this manuscript is now acceptable for publication, you may indicate that here to bypass the “Comments to the Author” section, enter your conflict of interest statement in the “Confidential to Editor” section, and submit your "Accept" recommendation.

Reviewer #1: All comments have been addressed

Reviewer #2: (No Response)

2. Is the manuscript technically sound, and do the data support the conclusions?

Reviewer #1: Yes

Reviewer #2: Partly

3. Has the statistical analysis been performed appropriately and rigorously? 

Reviewer #1: Yes

Reviewer #2: No

4. Have the authors made all data underlying the findings in their manuscript fully available?

Reviewer #1: Yes

Reviewer #2: Yes

5. Is the manuscript presented in an intelligible fashion and written in standard English?

Reviewer #1: Yes

Reviewer #2: Yes

6. Review Comments to the Author

Reviewer #1: Congratulontions on your article!

After the suggested changes it is clear and cohesive. Unfortonately missing data could not be included, what could have added more relevance.

The added references made the difference in the discussion.

Reviewer #2: Thanks for submitting the 1st revision of the manuscript. After careful consideration of the result of bivarite and multivariate analysis, and other supplementary materials, I think some additional revision is required to meet the adequate standard of a scientific manuscript. I suggested that authors should review some systematic review of IYCF practices, especially enablers and barriers of IYCF practices in the relevant context. The reflection of these systematic reviews is required throughout the whole manuscript. Authors need to do possible modification wherever possible otherwise need to mention as limitations. I am mentioning some of my specific observations below,

Title

Throughout the manuscript, especially the Result and Discussion, I do not find any special reflection of maternal factors on all IYCF practices. So, I suggest not to use maternal factors in the title, rather mentioning about general determinants or factors of IYCF in western hilly region of Nepal.

Abstract

Authors need to revise Abstract according to overall revision of manuscript.

Introduction

1. Line 83-89: Please omit the special importance of maternal factors and make it general. Focus on the rationale of IYCF exploration in the hilly area of Nepal.

Materials and methods

1. Sampling: Generally in Multistage sampling, PPS was done to select 1st stage cluster (municipality), and in second stage fixed number of sample is taken from each 1st stage cluster. This is self weighting sampling and having some statistical advantage. But authors did the opposite (missed to take a statistical opportunity). No matter what, please write, how many rural and urban municipalities are in the Syangja district?

2. Line 122-28: The details of maternal autonomy questionnaire need to incorporate in “Independent variables” section.

3. Maternal knowledge on feeding practice: How the low, medium and high knowledge is categorized? A combined score of maternal knowledge on different child feeding practice at different age may not have the causal temporal relationship with all IYCF practice. For example, knowledge of breastfeeding may not plausibly improve the dietary diversity practice. So, I suggest not to create a combined knowledge score, rather use them independently as separate independent variable whereas applicable in relevant analysis table.

4. Statistical analysis: Refer S6 (Bivariate analysis final) and also tell that the variable found significant in chi-square test included in bivariate and multivariate logistic regression.

5. Line 196-97: Drop the sentence “Multi-collinearity was assessed before performing logistic regression by Chi-square test”. Multi-collinearity is not problem here because a few independent variables found significant at first stage.

Results

1. Authors need to make some more categories of education for understanding the doge response relationship. Make at least 3 categories of consistent with Nepal DHS or other similar literature of Nepal. Why the informal education comes as a separate category? Primarily, I am suggesting 4 categories, i.e., illiterate, informal+primary, secondary, and higher secondary or above. Authors could explore different ordered categories. Also use same education category for all outcome variables’ table.

2. Authors need to use the frequency of ANC as category for understanding the doge response. For example, 0 ANC, 1-3 ANC, 4-6 ANC and 7+ANC. Existing literatures emphasized the relationship of ANC number and IYCF practices more.

3. Authors should try to use same independent variables wherever applicable for all the tables of S6 (Bivariate analysis final). In table 2 of S6, Sex of the child must be included.

4. Birth order of last child is an important determinant of most of IYCF practices. Number of child could be a proxy of birth order. The number of child should be re-categorized as 1, 2, 3 and 4+.

5. The order of Yes/No column of all tables of S6 should be unique.

6. Some important determinants of IYCF were not included in the analysis. For example, preterm birth, perceived size of child, lower birth order, parity, household food security, father’s education, maternal employment, exposure to media, family level health promoter’s visit for IYCF promotion and IYCF specific counseling and knowledge development (i.e., whether mother received counseling on timely initiation of breastfeeding, exclusive breastfeeding, timely initiation of complementary feeding, dietary diversity and age specific meal frequency; and whether mother developed the specific knowledge of specific IYCF practice). Not all of these indicators were collected in the survey, but some of their proxy can be used. Authors should mention the reasons for not collecting these important variables in the limitation section and also need to write about the way out using the proxy variables.

7. How “Food enough to eat for” variable used in the multivariate analysis. This variable is only applicable for if “Crop production” is Yes. “Crop production” is No will make pair-wise case deletion in the multivariate analysis. Authors could make a composite indicator named “Crop production and food security”. Indicator option: 1. Not produce crop or food not enough for 12 months, 2. Produced crop and food enough for 12 months

8. Maternal occupation should be categorized as standard order of maternal employment.

9. The effective sample size of exclusive breastfeeding analysis is low. Authors need to write in limitation section.

10. Line 232-33 in 1st revised version: 91.9% become duplicated.

11. Line 238-43: Write the applicable age range of child.

12. Table 4, 5, 6 and 7: Add p-value of Crude OR.

13. Table 4, 5, 6 and 7: Write uniquely Crude OR or Unadjusted OR, also Explanatory variable or Variable.

14. For Crude OR, Adjusted OR and 95% CI, use 1 digit decimal rounding; and for p-value, use 3 digit decimal rounding. Adequate rounding will help the readers to follow easily.

15. Table 6: The crude OR of Growth monitoring and Nutrition counseling need to recheck and corrected. This is not matching with the proportion of bivariate analysis of S6.

16. Table 7: The crude OR of PNC visit also need to recheck and corrected. This is also not matching with the proportion of bivariate analysis of S6.

17. Line 248-78: Table 4, 5, 6 and 7 are the subsequent analysis of S6. At the beginning of write-up, authors need to add an introductory paragraph noting the analysis sequence of S6 to Table 4, 5, 6 and 7.

Discussion

In discussion, the reflection of all IYCF papers conducted in Nepal must be included. I found some more IYCF papers of Nepal than authors referenced in the discussion. The reasons of similarity or dissimilarity of the result should be convincingly interpreted. Next to the Nepalese IYCF literatures, authors should consider the IYCF papers of South Asian countries, and then if required, need to consider the African, Latin American and East asian context. Authors should explore the recent systematic reviews of South Asian countries. Some of my specific comments of discussion are,

1. From my knowledge, there are sufficient contextual differences among the geographical regions of Nepal, and the context of this study is especially focused in western hilly area. If all region of Nepal is similar then there is no need of this separate study. So, in comparison and result interpretation the differential contextual issue always needs to keep in mind. The result interpretation and programmatic recommendation always need to be concentrated in western hilly area of Nepal.

2. Line 280-85: The sample size of currently breastfeeding and timely initiation of breastfeeding is not less. What is the programmatic implication of such high rate of breastfeeding in the hilly area of Nepal or Nepal in general? Is this normal secular trend? Is the current government and non-government policy and program is doing well? Is there any future programmatic recommendation? Review more relevant literatures and add logical insight.

3. Authors also need to discuss about the proportion of Initiation of complementary feeding, Minimum dietary diversity and Minimum acceptable diet. Authors also need to generate some programmatic recommendations.

4. Line 286-94: Compare the results with Nepal’s literatures first, then the country of South Asia, if unavailable then can be compared with other countries. What is the reason that nuclear family more likely to initiate breastfeeding on time? Did the Colombian study mention any reason?

5. Line 295-98: Lower than the national average may be only due the low sample size. Were the Cameroon and North West Ethiopia’s studies’ EBF prevalence also lower than their National EBF prevalence? If yes then this reference can be used, otherwise need to delete. The language also need to change, so that we can be understood the similarly of situation that the studies encountered similarly.

6. Line 314-325: Reduce the text. Drop the similar sentence. Inverse association need not to mention here. If mentioned then the reason also need to mention. “Mothers with higher autonomy received more advice about optimal breastfeeding while attending antenatal care.”—Is this really true? How is it? Add reference. Otherwise delete.

7. Line 326-33: Wealth quintile is most important factor found of timely initiation of complementary feeding in most of the similar literature. Why wealth quintile is not a factor for this target population? Authors need to try to explain.

8. Line 332-33: “Considering the sociocultural practices of Nepalese society, female children are introduced early with complementary feeding.”---Need correction

9. Line 334-51: The non-association of wealth quintile needs to explain. The negative association of maternal autonomy of Benin and Nigeria need not to mention.

10. Line 299-351: In addition to the comparing result and interpretation, some practical and feasible way forward recommendations for nutrition policy, programming and intervention for Nepal or hilly area of Nepal are required. Authors need to include some specific way forward recommendations.

Conclusion

1. The summary of specific recommendations of discussion need to include in Conclusion section.

2. Finally, after the revision, I suggest a 3rd person’s language edit to improve the standard of language.

7. PLOS authors have the option to publish the peer review history of their article (what does this mean?). If published, this will include your full peer review and any attached files.

Reviewer #1: No

Reviewer #2: **Yes: **Rashidul Azad, Research Investigator, International Centre for Diarrhoeal Disease Research, Bangladesh (icddr,b)

---

## [Author Response · Author response to Decision Letter 1]

28 Jun 2021

Reviewer #1: Congratulontions on your article!

After the suggested changes it is clear and cohesive. Unfortonately missing data could not be included, what could have added more relevance.

The added references made the difference in the discussion.

Reviewer #2: Thanks for submitting the 1st revision of the manuscript. After careful consideration of the result of bivarite and multivariate analysis, and other supplementary materials, I think some additional revision is required to meet the adequate standard of a scientific manuscript. I suggested that authors should review some systematic review of IYCF practices, especially enablers and barriers of IYCF practices in the relevant context. The reflection of these systematic reviews is required throughout the whole manuscript. Authors need to do possible modification wherever possible otherwise need to mention as limitations. I am mentioning some of my specific observations below,

Title

Throughout the manuscript, especially the Result and Discussion, I do not find any special reflection of maternal factors on all IYCF practices. So, I suggest not to use maternal factors in the title, rather mentioning about general determinants or factors of IYCF in western hilly region of Nepal.

Response: Thank you for your comments. We have modified the title accordingly.

Abstract

Authors need to revise Abstract according to overall revision of manuscript.

Response: We have revised the abstract according to revised manuscript. 

Introduction

1. Line 83-89: Please omit the special importance of maternal factors and make it general. Focus on the rationale of IYCF exploration in the hilly area of Nepal.

Materials and methods

Response: Thank you for the comment. We have modified the introduction accordingly.

1. Sampling: Generally in Multistage sampling, PPS was done to select 1st stage cluster (municipality), and in second stage fixed number of sample is taken from each 1st stage cluster. This is self weighting sampling and having some statistical advantage. But authors did the opposite (missed to take a statistical opportunity). No matter what, please write, how many rural and urban municipalities are in the Syangja district?

Response: Thank you very much for the comments. We have already mentioned in methodology that Syangja district has 5 urban municipalities and 6 rural municipalities. 

2. Line 122-28: The details of maternal autonomy questionnaire need to incorporate in “Independent variables” section.

Response : Thank you very much for your comments. We incorporated it in independent variables section. 

3. Maternal knowledge on feeding practice: How the low, medium and high knowledge is categorized? A combined score of maternal knowledge on different child feeding practice at different age may not have the causal temporal relationship with all IYCF practice. For example, knowledge of breastfeeding may not plausibly improve the dietary diversity practice. So, I suggest not to create a combined knowledge score, rather use them independently as separate independent variable whereas applicable in relevant analysis table.

Response: Thank you very much for your comments. We have incorporated your suggestion. 

4. Statistical analysis: Refer S6 (Bivariate analysis final) and also tell that the variable found significant in chi-square test included in bivariate and multivariate logistic regression.

Response: Thank you for your comments. We have mentioned this in statistical analysis section. 

5. Line 196-97: Drop the sentence “Multi-collinearity was assessed before performing logistic regression by Chi-square test”. Multi-collinearity is not problem here because a few independent variables found significant at first stage.

Response: Thank you for your comment. We have addressed this now. 

Results

1. Authors need to make some more categories of education for understanding the doge response relationship. Make at least 3 categories of consistent with Nepal DHS or other similar literature of Nepal. Why the informal education comes as a separate category? Primarily, I am suggesting 4 categories, i.e., illiterate, informal+primary, secondary, and higher secondary or above. Authors could explore different ordered categories. Also use same education category for all outcome variables’ table.

Response: Thank you for your comments. We have addressed the comment. We have also used same categories of education in bivariate and multivariate analysis. 

2. Authors need to use the frequency of ANC as category for understanding the doge response. For example, 0 ANC, 1-3 ANC, 4-6 ANC and 7+ANC. Existing literatures emphasized the relationship of ANC number and IYCF practices more.

Response: Thank you so much for your comments. We have categorized the variable as per your suggestion and used the same in bivariate and multivariate analysis. 

3. Authors should try to use same independent variables wherever applicable for all the tables of S6 (Bivariate analysis final). In table 2 of S6, Sex of the child must be included.

Response: Thank you for your comment. It has been addressed.

4. Birth order of last child is an important determinant of most of IYCF practices. Number of child could be a proxy of birth order. The number of child should be re-categorized as 1, 2, 3 and 4+.

Response: Thank you for your comment. It has been re-categorized.

5. The order of Yes/No column of all tables of S6 should be unique.

Response: Thank you so much for your comment. It has been addressed.

6. Some important determinants of IYCF were not included in the analysis. For example, preterm birth, perceived size of child, lower birth order, parity, household food security, father’s education, maternal employment, exposure to media, family level health promoter’s visit for IYCF promotion and IYCF specific counseling and knowledge development (i.e., whether mother received counseling on timely initiation of breastfeeding, exclusive breastfeeding, timely initiation of complementary feeding, dietary diversity and age specific meal frequency; and whether mother developed the specific knowledge of specific IYCF practice). Not all of these indicators were collected in the survey, but some of their proxy can be used. Authors should mention the reasons for not collecting these important variables in the limitation section and also need to write about the way out using the proxy variables.

Response: Thank you for your comments. We have addressed this in limitations.

7. How “Food enough to eat for” variable used in the multivariate analysis. This variable is only applicable for if “Crop production” is Yes. “Crop production” is No will make pair-wise case deletion in the multivariate analysis. Authors could make a composite indicator named “Crop production and food security”. Indicator option: 1. Not produce crop or food not enough for 12 months, 2. Produced crop and food enough for 12 months

Response: Thank you so much for your comments. This has been addressed accordingly.

8. Maternal occupation should be categorized as standard order of maternal employment.

Response: Thank you so much for your comment. We have classified the occupation accordingly. 

9. The effective sample size of exclusive breastfeeding analysis is low. Authors need to write in limitation section.

Response: Thank you for your comment. We have mentioned it as limitation.

10. Line 232-33 in 1st revised version: 91.9% become duplicated.

Response: Thank you for your comment. This has been corrected.

11. Line 238-43: Write the applicable age range of child.

Response: Thank you for your comment. The standard age categories to assess infant feeding practices was used 

12. Table 4, 5, 6 and 7: Add p-value of Crude OR.

Response: Thank you for your comment. We have added it.

13. Table 4, 5, 6 and 7: Write uniquely Crude OR or Unadjusted OR, also Explanatory variable or Variable.

Response: Thank you for your comments. We have now written it accordingly.

14. For Crude OR, Adjusted OR and 95% CI, use 1 digit decimal rounding; and for p-value, use 3 digit decimal rounding. Adequate rounding will help the readers to follow easily.

Response: Thank you for the comment. We have corrected it.

15. Table 6: The crude OR of Growth monitoring and Nutrition counseling need to recheck and corrected. This is not matching with the proportion of bivariate analysis of S6.

Response: Thank you for your comment. We have addressed it.

16. Table 7: The crude OR of PNC visit also need to recheck and corrected. This is also not matching with the proportion of bivariate analysis of S6.

Response: Thank you for your comment. We have addressed it.

17. Line 248-78: Table 4, 5, 6 and 7 are the subsequent analysis of S6. At the beginning of write-up, authors need to add an introductory paragraph noting the analysis sequence of S6 to Table 4, 5, 6 and 7.

Response: Thank you for your comment. We have added a line in the results section.

Discussion

In discussion, the reflection of all IYCF papers conducted in Nepal must be included. I found some more IYCF papers of Nepal than authors referenced in the discussion. The reasons of similarity or dissimilarity of the result should be convincingly interpreted. Next to the Nepalese IYCF literatures, authors should consider the IYCF papers of South Asian countries, and then if required, need to consider the African, Latin American and East asian context. Authors should explore the recent systematic reviews of South Asian countries. Some of my specific comments of discussion are,

1. From my knowledge, there are sufficient contextual differences among the geographical regions of Nepal, and the context of this study is especially focused in western hilly area. If all region of Nepal is similar then there is no need of this separate study. So, in comparison and result interpretation the differential contextual issue always needs to keep in mind. The result interpretation and programmatic recommendation always need to be concentrated in western hilly area of Nepal.

Response: Thank you for your comment. We have modified it accordingly.

2. Line 280-85: The sample size of currently breastfeeding and timely initiation of breastfeeding is not less. What is the programmatic implication of such high rate of breastfeeding in the hilly area of Nepal or Nepal in general? Is this normal secular trend? Is the current government and non-government policy and program is doing well? Is there any future programmatic recommendation? Review more relevant literatures and add logical insight.

Response: Thank you for your comment. We have modified it accordingly 

3. Authors also need to discuss about the proportion of Initiation of complementary feeding, Minimum dietary diversity and Minimum acceptable diet. Authors also need to generate some programmatic recommendations.

Response: Thank you for your comment. We have modified it accordingly 

4. Line 286-94: Compare the results with Nepal’s literatures first, then the country of South Asia, if unavailable then can be compared with other countries. What is the reason that nuclear family more likely to initiate breastfeeding on time? Did the Colombian study mention any reason?

Response: Thank you for your comment. We have modified it accordingly

5. Line 295-98: Lower than the national average may be only due the low sample size. Were the Cameroon and North West Ethiopia’s studies’ EBF prevalence also lower than their National EBF prevalence? If yes then this reference can be used, otherwise need to delete. The language also need to change, so that we can be understood the similarly of situation that the studies encountered similarly.

Response: Thank you for your comment. We have modified it accordingly

6. Line 314-325: Reduce the text. Drop the similar sentence. Inverse association need not to mention here. If mentioned then the reason also need to mention. “Mothers with higher autonomy received more advice about optimal breastfeeding while attending antenatal care.”—Is this really true? How is it? Add reference. Otherwise delete.

Response: Thank you for your comment. We have modified it accordingly

7. Line 326-33: Wealth quintile is most important factor found of timely initiation of complementary feeding in most of the similar literature. Why wealth quintile is not a factor for this target population? Authors need to try to explain.

Response: Thank you for your comment. We have modified it accordingly

8. Line 332-33: “Considering the sociocultural practices of Nepalese society, female children are introduced early with complementary feeding.”---Need correction

Response: Thank you for your comment. We have corrected it.

9. Line 334-51: The non-association of wealth quintile needs to explain. The negative association of maternal autonomy of Benin and Nigeria need not to mention.

Response: Thank you for your comment. We have modified it accordingly

10. Line 299-351: In addition to the comparing result and interpretation, some practical and feasible way forward recommendations for nutrition policy, programming and intervention for Nepal or hilly area of Nepal are required. Authors need to include some specific way forward recommendations.

Response: Thank you for your comment. We have modified it accordingly

Conclusion

1. The summary of specific recommendations of discussion need to include in Conclusion section.

Response: Thank you for your comment. We have modified it accordingly

2. Finally, after the revision, I suggest a 3rd person’s language edit to improve the standard of language.

Response: Thank you for your comment. Language editing has been performed.

---

## [Decision Letter · Decision Letter 2]

28 Sep 2021

PONE-D-21-02451R2

Infant and young child feeding practices and its associated factors among mothers of under two years children in a western hilly region of Nepal

PLOS ONE

Dear Dr. Pradhan,

Thank you for submitting your manuscript to PLOS ONE. After careful consideration, we feel that it has merit but does not fully meet PLOS ONE’s publication criteria as it currently stands. Therefore, we invite you to submit a revised version of the manuscript that addresses the points raised during the review process.

A few minor edits have been requested by the reviewers to further improve the language of your manuscript. Please see the attached reviewer comments for specific locations in the manuscript that require further attention. Thank you for your continued efforts on this exciting paper on early life feeding.

We look forward to receiving your revised manuscript.

Kind regards,

Corrie Whisner

Academic Editor

PLOS ONE

Journal Requirements:

Reviewers' comments:

Reviewer's Responses to Questions

**Comments to the Author**

1. If the authors have adequately addressed your comments raised in a previous round of review and you feel that this manuscript is now acceptable for publication, you may indicate that here to bypass the “Comments to the Author” section, enter your conflict of interest statement in the “Confidential to Editor” section, and submit your "Accept" recommendation.

Reviewer #1: All comments have been addressed

Reviewer #2: (No Response)

2. Is the manuscript technically sound, and do the data support the conclusions?

Reviewer #1: Yes

Reviewer #2: Yes

3. Has the statistical analysis been performed appropriately and rigorously? 

Reviewer #1: Yes

Reviewer #2: Yes

4. Have the authors made all data underlying the findings in their manuscript fully available?

Reviewer #1: Yes

Reviewer #2: Yes

5. Is the manuscript presented in an intelligible fashion and written in standard English?

Reviewer #1: Yes

Reviewer #2: Yes

6. Review Comments to the Author

Reviewer #1: Congratulations on your paper!

Feeding pratices is a very important theme, specially in low income countries, and we should have more studies in this field.

You did some language improvements. Your text is more concise and your writting is clearer. Introduction is straight foward. Methods are precise. Results are clean and related to the tables. Discussion has relevant references and more cohesive.

The only typing error is in line 304, after ref 33. Probably there is missing a comma or a preposition.

Reviewer #2: Thanks authors for submitting the 2nd revised version. Congratulations for your hard work. I have few minor comments need to address before publication. In addition to that an external person’s final language checking of the unspotted errors would improve the overall language quality of manuscript.

Introduction

1. Line 55-56: “Appropriate infant and young child feeding practices help to prevent almost of all under-five deaths.” --- Please add citation with appropriate reference.

2. Line 65-66: “The government of Nepal has developed and implemented different acts, policies, strategies, and programs to improve infant feeding practices.” --- Please add citation with appropriate reference.

3. Line 75-67: “Maternal factors such as decision-making capacity, education, knowledge, and maternal health services utilization are important factors for child feeding practices.” --- Please add citation with appropriate reference.

4. Line 81-83: Write in simple language mentioning the prevalence of plain and hilly area. Your sentence structure sounds that prevalence of hilly area is lower than plain area!

Materials and methods

1. Line 96-101: Difficult to follow. Write hierarchical and simple language.

2. Line 171-174: Add score of reliability of scale here not in Data Quality control section.

Results

1. Line 214: “26.20 ± 4.148”--- Make 1 decimal point. Do it throughout manuscript except p-value.

2. Line 229-30: “Less than half (47.6%) of children were exclusively breastfed.” --- Mention the age range of denominator of this proportion for better understanding.

3. Line 230-31: “More than half (53.3%) of children were initiated with complementary feeding timely. Similarly, nearly half (49.4%) of children were fed according to recommended.” --- Mention the age range of denominator of this proportion for better understanding.

4. “Instutional” spelling should be corrected in Table 6 and Table 7.

Discussion

1. Line 284-86: “This would suggest that more efforts are needed on improving the quality of maternal and neonatal health care services at health institutions.”--- What kind of effort can be given to initiate breastfeeding within one hour after delivery?

2. Line 288-88: “Mothers who were from nuclear family were more likely to initiate breastfeeding on time.”--- Please explain why

3. Line 294-96: “This variation might be due to the socio-economic status of the participants, access to a health facility, and sample size compared to this study.”--- This effective sample size of EBF is 103. This small sample size is incapable to capture variation socio-economic status and access to a health facility. So, emphasize the small sample size as only limiting factor and discard socio-conomic status and access to the health facility.

4. Line 298-302: By definition of EBF is considered 24 hours before interview. How these facility level speculations are linked with EBF within last 24 hours?

5. Line 296-308: The factor determination of EBF is seriously flawed by the small sample size. So, extended discussion and stronger recommendation is not applicable. For EBF, I only prefer mentioning the result and telling the consistency of literature in similar settings. IF inconsistent findings found then tell about the small sample size as a probable reason.

6. Line 343-45: Recommendation of further research need to add with the higher sample size (especially for the narrower age range) with other applicable robust designs.

7. Line 346-47: “In this study, only six indicators were used to assess the feeding practices of children.”--- Is there any more WHO indicators to assess the IYCF? Mention about the independent variables that you missed in the questionnaire.

Conclusion

1. Line 362-64: “We also need to address the service quality and utilization of maternal health services which is considered to be an important determinant of child feeding practices.”--- In this study service quality is not measured.

7. PLOS authors have the option to publish the peer review history of their article (what does this mean?). If published, this will include your full peer review and any attached files.

Reviewer #1: No

Reviewer #2: **Yes: **Md. Rashidul Azad,

Research Investigator,

icddr,b, Bnagladesh

---

## [Author Response · Author response to Decision Letter 2]

6 Nov 2021

Introduction

1. Line 55-56: “Appropriate infant and young child feeding practices help to prevent almost of all under-five deaths.” --- Please add citation with appropriate reference.

Response – Thank you so much for your comment. We added the citation. 

2. Line 65-66: “The government of Nepal has developed and implemented different acts, policies, strategies, and programs to improve infant feeding practices.” --- Please add citation with appropriate reference.

Response – Thank you so much for your comment. We added the citation. 

3. Line 75-67: “Maternal factors such as decision-making capacity, education, knowledge, and maternal health services utilization are important factors for child feeding practices.” --- Please add citation with appropriate reference.

Response – Thank you so much for your comment. We added the citation. 

4. Line 81-83: Write in simple language mentioning the prevalence of plain and hilly area. Your sentence structure sounds that prevalence of hilly area is lower than plain area!

Response – Thank you so much for your comments. We have written it in simple language.

Materials and methods

1. Line 96-101: Difficult to follow. Write hierarchical and simple language.

Response – Thank you so much for your comments. We have now made it clear. 

2. Line 171-174: Add score of reliability of scale here not in Data Quality control section.

Results

Response – Thank you so much for your comments. We added reliability score. 

1. Line 214: “26.20 ± 4.148”--- Make 1 decimal point. Do it throughout manuscript except p-value.

Response – Thank you so much for your comments. We have corrected it.

2. Line 229-30: “Less than half (47.6%) of children were exclusively breastfed.” --- Mention the age range of denominator of this proportion for better understanding.

Response – Thank you so much for your comments. We have added the age range. 

3. Line 230-31: “More than half (53.3%) of children were initiated with complementary feeding timely. Similarly, nearly half (49.4%) of children were fed according to recommended.” --- Mention the age range of denominator of this proportion for better understanding.

Response – Thank you so much for your comments. We added the age range. 

4. “Instutional” spelling should be corrected in Table 6 and Table 7.

Response – Thank you so much for your comments. We have corrected it. 

Discussion

1. Line 284-86: “This would suggest that more efforts are needed on improving the quality of maternal and neonatal health care services at health institutions.”--- What kind of effort can be given to initiate breastfeeding within one hour after delivery?

Response – Thank you so much for your comments. We have added the points in discussion. 

2. Line 288-88: “Mothers who were from nuclear family were more likely to initiate breastfeeding on time.”--- Please explain why

Response – Thank you so much for your comments. We have added the possible reason. 

3. Line 294-96: “This variation might be due to the socio-economic status of the participants, access to a health facility, and sample size compared to this study.”--- This effective sample size of EBF is 103. This small sample size is incapable to capture variation socio-economic status and access to a health facility. So, emphasize the small sample size as only limiting factor and discard socio-conomic status and access to the health facility.

Response – Thank you so much for your comments. We have corrected it.

4. Line 298-302: By definition of EBF is considered 24 hours before interview. How these facility level speculations are linked with EBF within last 24 hours?

Response – Thank you so much for your comments. We have corrected it. 

5. Line 296-308: The factor determination of EBF is seriously flawed by the small sample size. So, extended discussion and stronger recommendation is not applicable. For EBF, I only prefer mentioning the result and telling the consistency of literature in similar settings. IF inconsistent findings found then tell about the small sample size as a probable reason.

Response – Thank you so much for your comments. We have corrected it.

6. Line 343-45: Recommendation of further research need to add with the higher sample size (especially for the narrower age range) with other applicable robust designs.

Response – Thank you so much for your comments. We have corrected it. 

7. Line 346-47: “In this study, only six indicators were used to assess the feeding practices of children.”--- Is there any more WHO indicators to assess the IYCF? Mention about the independent variables that you missed in the questionnaire.

Response – Thank you so much for your comments. We have corrected it.

Conclusion

1. Line 362-64: “We also need to address the service quality and utilization of maternal health services which is considered to be an important determinant of child feeding practices.”--- In this study service quality is not measured.

Response – Thank you so much for your comments. We have removed the sentence on service quality.

---

## [Editor Report · Decision Letter 3]

1 Dec 2021

Infant and young child feeding practices and its associated factors among mothers of under two years children in a western hilly region of Nepal

PONE-D-21-02451R3

Dear Dr. Pradhan,

We’re pleased to inform you that your manuscript has been judged scientifically suitable for publication and will be formally accepted for publication once it meets all outstanding technical requirements.

Kind regards,

Corrie Whisner

Academic Editor

PLOS ONE
---

## [Editor Report · Acceptance letter]

6 Dec 2021

PONE-D-21-02451R3 

Infant and young child feeding practices and its associated factors among mothers of under two years children in a western hilly region of Nepal 

Dear Dr. Pradhan:

I'm pleased to inform you that your manuscript has been deemed suitable for publication in PLOS ONE. Congratulations! Your manuscript is now with our production department. 

Kind regards, 

on behalf of

Dr. Corrie Whisner 

Academic Editor

PLOS ONE